# Effects of Follicular Fluid on Physiological Characteristics and Differentiation of Fallopian Tube Epithelial Cells Implicating for Ovarian Cancer Pathogenesis

**DOI:** 10.3390/ijms241210154

**Published:** 2023-06-15

**Authors:** Maobi Zhu, Na Wang, Sha Wang, Yao Wang, Xiawen Yang, Jianglin Fan, Yajie Chen

**Affiliations:** 1Guangdong Provincial Key Laboratory of Large Animal Models for Biomedicine, South China Institute of Large Animal Models for Biomedicine, Wuyi University, Jiangmen 529000, China; maobizhu@outlook.com (M.Z.); wangna12240801@163.com (N.W.); wangsha1551@163.com (S.W.); wangyao_1125@hotmail.com (Y.W.); yangxiawen@outlook.com (X.Y.); 2Department of Molecular Pathology, Interdisciplinary Graduate School of Medicine, University of Yamanashi, Yamanashi 409-3898, Japan

**Keywords:** fallopian tube, ciliogenesis, follicular fluid, organoid culture, cell injury, estrogen

## Abstract

The fallopian tube (FT) is an important reproductive organ in females. Ample evidence suggests that the distal end of FT is the original site of high-grade serous ovarian carcinoma (HGSC). FT may suffer from repeated injury and repair stimulated by follicular fluid (FF); however, this hypothesis has not been examined. In fact, the molecular mechanism of homeostasis, differentiation, and the transformation of fallopian tube epithelial cells (FTECs) resulting from the stimulation of FF are still enigmatic. In this study, we examined the effects of FF along with factors present in the FF on a variety of FTEC models, including primary cell culture, ALI (air–liquid interface) culture, and 3D organ spheroid culture. We found that FF plays a similar role to estrogen in promoting cell differentiation and organoid formation. Moreover, FF significantly promotes cell proliferation and induces cell injury and apoptosis in high concentrations. These observations may help us to investigate the mechanisms of the initiation of HGSC.

## 1. Introduction

The fallopian tube (FT) is an important reproductive organ in females. From the distal end to the uterus, the fallopian tube can be anatomically divided into three parts: the fimbria, ampulla, and isthmus. The lumen of the FT is covered by simple columnar epithelial cells, including basal cells and secretory and ciliated cells, and these cells play critical roles in conditioning the FT lumen surface for efficient reproduction [1]. Because of the menstrual cycle in humans, the streams of ovarian steroid hormones such as progesterone and estrogen fluctuate dramatically, and the fallopian tube epithelial cells (FTECs) usually undergo periodic changes [2]. In our previous study, we demonstrated that estradiol (E2) promoted ciliogenesis and maintained homeostasis of the fallopian tube epithelium [3]. In addition to the hormones, the fallopian tube, especially the fimbria part, is subjected to the stimulation of follicular fluid (FF) released by cyclic ovulation. This local microenvironmental stress induced by ovulation is considered a risk factor for high-grade serous ovarian cancer (HGSC) [2,4,5,6,7]. Several studies showed that the HGSC may be originated from the fimbria [8,9,10]. However, it is still unknown how neoplastic lesions of the FT are associated with the development of HGSC. Therefore, disclosing the molecular mechanisms of fimbria long-term renewal and integrity is of critical importance for understanding the influence of FF.

FF is derived either from circulating proteins or secreted by mural and cumulus granulosa cells [11,12], which plays a crucial role in determining the quality of the oocytes and affecting fertilization and embryonic development in mammals [13,14,15,16]. FF is a complex liquid primarily composed of steroid hormones and other biological materials such as growth factors, cytokines, reactive oxygen species (ROS), prostanoids, and proteolytic enzymes [17]. Mature follicles contain as much as 1000-fold higher levels of estrogen and progesterone than the blood [18,19]. Among these factors in FF, many of them are considered carcinogenetic such as ROS, insulin-like growth factor (IGF), and some cytokines, because they exert various functions on cell transformation, DNA double-strand breaks, stemness activation, and clonal expansion [5,20]. Because the components in FF are complicated, the exact roles of FF on the tubal epithelium are still not fully clarified.

To investigate the FF functions, we established the culture system for the stable expansion of basal stem cells from FT, as well as that for differentiation and long-term culture in ALI conditions [21]. By using this stable high-fidelity model of cultured FTECs, we further elucidate the functional roles of the FF on FTECs, including cell proliferation, differentiation, and cellular injury. These results may help us to investigate the mechanism of the initiation of HGSC.

## 2. Results

### 2.1. The Effect of Porcine Follicular Fluid (PFF) on the Primary FTECs

To investigate the biological effects of PFF and the hormones on FTECs, we first analyzed p73 expression and the proliferation rate by EdU assay in FTECs. FTECs were passaged and incubated in the presence of E2, progesterone (P4), and PFF for 24 h, respectively. A total of 5 μM Edu was added into each group and further incubated for another 24 h. Compared to the control, E2, and P4 groups, treatment with 1% and 5% PFF in FTECs led to a significant increase in p73 expression, a progenitor maker expressed in FTECs (Figure 1A,B). In addition, EdU-positive cells in the group of 5% PFF were significantly greater than other groups (Figure 1A,C), suggesting that PFF may enhance cellular proliferation. These results indicate that FF may increase the growth rate of primary FTECs, but generally, it does not affect precursor cells and retain the differentiation potential of FTECs.

### 2.2. The Effect of PFF on the Organoid Formation in 3D Matrigel Culture

It is known that FTECs can form organoids and recapitulate the mucosal fold architecture for long-term culture [22]. To obtain insight into the effects of the PFF on the fallopian tube organoids, we carried out a 3D culture of FTECs by using Matrigel. Briefly, expanded FTECs were suspended with 100 μL Matrigel and seeded on a 24-well plate. The organoids were incubated in a basal medium as a control, whereas other groups were incubated in the presence of E2, P4, and PFF. In general, small round clusters of cells were visible after 48 h. In the presence of E2 and various concentrations of PFF, FTECs developed into rapidly expanding spheres of a round cystic phenotype by 5 days post-seeding (Figure 2). Compared to the control group, the concentrations ranged from 1% to 20% PFF apparently increased the number as well as the diameter of organoid spheres from 5 days, similar to E2. The size of the organoids in the group treated by E2 and PFF reached a peak by 2 weeks (Figure 2). Subsequently, we carried out another experiment in which FTECs were incubated with E2, P4, and 1% PFF for 6, 10, 15, and 19 days. The number and size of spheres with diameters greater than 50 μm were counted. Compared with the control, P4 slightly increased the number and size of organoid spheres, by not as much as E2 and PFF (Figure 3A,B). All E2, P4, and PFF showed beneficial effects for the formation of organoid spheres, but E2 exhibited the strongest effect (Figure 3A,B).

### 2.3. The Effect of PFF on the Differentiation of FTECs in ALI Culture

Because of the complexity of FF, it is still unclear what the overall regulatory effects of FF exert on the differentiation of FTECs. Therefore, we examined the effect of PFF on the differentiation of FTECs by using an ALI culture model. Consistent with our assumption, PFF indeed promoted ciliogenesis in a similar manner to E2. In contrast, treatment with P4 led to very few numbers of MCCs formation (Figure 4A,B). These results indicate that E2 is the major component in PFF that induces ciliogenesis. Moreover, we also analyzed cell proliferation and renewal in the long-term ALI culture. We added EdU to the media for 48 h on the ALI day 10 after FTECs differentiation induced by E2, P4, and PFF, respectively. The presence of 1% and 5% PFF showed a relatively higher proportion of EdU-positive cells than controls. EdU-positivity appears to be proportional to the number of ciliated cells. We supposed that E2 and PFF groups showed higher EdU positive cells because aging differentiated cells may increase, and the precursor cells divide to compensate for the aging cells (Figure 4A,C). We next analyzed the expression of Foxj1 and Pax8 expression in FTECs after being cultured without and with E2, P4, and 1% and 5% PFF, respectively. Compared to the control, treatment with E2, P4, and 1% and 5% PFF significantly increased Foxj1 expression, a ciliated cell maker expressed in FTECs. Moreover, P4 and 1% and 5% PFF also apparently promoted the expression of Pax8, a marker of secretory cells in FTECs (Figure 4D).

FF also contains many factors that modulate oocyte maturation and ovulation, such as various growth factors and cytokines, in addition to hormones. To elucidate the effect of these factors on the differentiation of FTECs, we examined the differentiation status of FTECs induced by several cytokines. For this purpose, FTECs were incubated with IL-1β, IL-6, TNFα, and IGF-1 in ALI conditions for 10 days, but we found that these factors alone failed to induce ciliogenesis compared with E2 (Figure 5A,C). When FTECs were treated by IL-1β, IL-6, TNFα, and IGF-1 in the presence of E2, only TNFα and estrogen exhibited a weak synergistic effect on ciliogenesis (Figure 5B,D). This observation suggests that E2 is indispensable for ciliogenesis, which is consistent with our previous study [3].

### 2.4. Simulating the Direct Stimulation of the Oviduct Epithelium by FF

Among the complex components in FF, ROS were considered to cause tissue injury and DNA double-strand breaks on the epithelium of fallopian tube fimbria and were regarded as mutagen in FF [23,24]. To explore whether FF indeed induces DNA damage, we performed TUNEL staining to detect DNA double-strand breaks. To mimic a situation that the fimbria is directly immersed and stimulated by a high concentration of FF during ovulation, we added 50% and 100% PFF into top chambers to simulate differentiated FTECs that were cultured and induced by E2 at ALI for 15 days. Treatment with either 50% PFF or 100% PFF for 30 min led to the appearance of TUNEL-positive cells, but the TUNEL-positive rate caused by 50% PFF was significantly lower than 100% PFF (Figure 6A), indicating that high concentrations of FF caused more serious damage to cells. Next, we wanted to test whether there was a time dependence of FF on damaging stimulation of the fallopian tubes. When fully differentiated FTECs were immersed into 100% PFF stimulation for 2.5 h, 7 h, and 12 h, the highest TUNEL positive rate at 2.5 h and a gradual decrease at 7 h and 12 h (Figure 6B). The result seems to be at odds with the actual expectation. We supposed that longer incubation with FF increases cellular death, and these dead cells may be detached and washed out during the staining process.

## 3. Discussion

In this study, we established three FTECs culture models, including 2D expanding culture, 3D, and ALI organoid culture model [21]. We extensively investigated the effects of FF on the physiological functions, differentiation, and transformation of FTECs in different stages by using a suitable culture model. Among these models, the ALI culture model has been introduced to FTECs, and it provides us with an excellent model to examine the function of FTECs [25,26]. In a previous study, we improved this ALI culture model, which maintains the morphological and functional properties of cells in vivo with an appropriate proportion of each cell type [21]. This model allows us to assess the physiological responses of specific cell subpopulations and enables us to understand better the mechanism of tumor initiation and progression in FTECs. Based on this model, we can recapture the direct stimulation of the oviduct epithelium by FF during ovulation.

Several previous reports have shown that serum levels of ovarian (mainly E2 and P4) regulate oviduct physiology, including the development of oocyte and embryo and differentiation of FTECs. In a previous study, we uncovered the role of the estrogen pathway in the differentiation of FTECs into MCCs [3]. Besides these cyclic changes in serum hormones affecting the FTECs, the apical compartment of the oviduct epithelium is temporarily exposed to FF, which contains much higher levels of the hormone than serum. FF contains a variety of hormones [2]. The concentration of E2 and P4 in FF varies significantly according to the size of follicles and menstrual cycle [27,28]. FF contains much higher levels of E2 than serum (up to 200 ng/mL) [18,29]. It is reported that such a high concentration of E2 could promote an inflammation-like reaction by increasing the levels of ROS in oviduct epithelial cells [30]. In this study, we first evaluated the long-term effects of residual FF on the FTECs. In this condition, we are mainly concerned with how the low-concentration FF and its related components (physiological concentrations.) regulate FTECs. FTECs are continuously stimulated by low concentrations of FF, increasing the growth rate of primary FTECs, but generally, it has no effect on precursor cells and retains the original differentiation potential of FTECs. FF contains many complex components, including some growth factors that may promote cell proliferation. On the other hand, FF plays a similar role to estrogen in promoting the organoid spheroid formation and differentiation of ciliated epithelium. However, other factors (IL-1β, IL-6, TNFα, IGF-1) present in FF do not have similar functions. Therefore, we consider that hormones, especially estrogen, play an important role in regulating the differentiation of the fallopian tube epithelium.

Numerous studies have shown that repeated stimulation of ovarian FF to the tubal end during ovulation will cause malignant mutation of FTECs, which contributes to the origin of HGSC [2,4,6,7,24]. During ovulation, fimbria must suffer a high concentration of hormones, growth factors, cytokines and interleukins, and ROS in a very short duration. We found that immersing differentiated FTECs into 100% PFF resulted in DNA damage. Undiluted or less diluted FF is highly toxic to fimbrial epithelial cells and tissue [6]. High levels of ROS and numerous inflammation factors in ovulatory FF induce DNA DSBs in the FTECs and accumulating DNA mutations, which confers a range of cell transformation phenotypes, including stemness activation and clonal expansion [6,23,31]. On the other hand, the presence of very high concentrations of hormones, especially estrogen, in the FF must also have some negative effects on the FTECs during ovulation. In another study, the exposure of FTECs to FF concentrations of E2 promoted transcription of IL6, IL8, and PTGS2, which triggered inflammatory and DNA damage responses in FTECs [29]. Considering FF contains much higher levels of E2 than serum, E2 is a potential candidate to promote an inflammation-like reaction by increasing the levels of ROS, resulting in DNA damage in FTE. Repeated tissue damage, repair, and renewal are the most common sites of oncogenesis. We speculate that FTECs themselves may repair DNA double-strand breaks caused by FF because this is necessary for FTECs to maintain homeostasis and normal physiological functions. However, when DNA mutation occurs at critical sites (such as P53, etc.) and fails to repair it, it results in malignant transformation in FTECs. Overall, the effects of FF on the oviduct epithelium during ovulation are extensive, not only involving complex factors but also extremely high concentrations of hormones.

In the present study, we employed a variety of cell models, including primary cell culture, ALI culture, and 3D organ spheroid culture, to extensively evaluate the effects of FF on the tubal epithelium. Understanding the process of ovulation-induced changes in FTECs enables us to understand the mechanism of tumor initiation and progression in FTECs better, which leads to HGSC.

## 4. Materials and Methods

### 4.1. Isolation and Culture of FTECs

Porcine fallopian tubes were collected from healthy sows at a local slaughterhouse in a sterile condition. The FTs were briefly washed with phosphate-buffered saline (PBS, pH 7.4) and opened longitudinally to expose the mucosal folds after removing adipose and connective tissue. Then, the whole FT was incubated with the digestion medium containing 100 U/mL collagenase type IV (17104-019, Thermo, Waltham, MA, USA) for 90 min at 37 °C. After incubation, the epithelial surface was scraped, and the cell suspension was collected. It was digested for another 15 min with an appropriate amount of Dnase I to make sure that the FTECs were totally released for the sticky clusters. Finally, the cell suspension was centrifugated at 1000 rpm for 5 min to collect cell pellets. FTECs were cultured according to our previous study [3]. We used basal medium consisting of DMEM/Ham’s F-12 medium (#042-30795, Wako Chemicals Com., Osaka, Japan) supplemented with 1% GlutaMAX (#35050061, Thermo), 2% B27 (#17504-044, Thermo), 1 mM nicotinamide (#72340, Sigma-Aldrich, St. Louis, MO, USA), 0.5 µM TGF β receptor kinase inhibitor IV (#SB431542, Wako), and 2 ng/mL human EGF (PHG0311, Thermo). The collected cells were resuspended in the basal media containing 10% FBS, followed by seeding onto a plastic dish, and then incubated at 37 °C in a 5% CO_2_-conditioned and humidified incubator. After the cells were attached to the dish, the culture medium was replaced with a serum-free medium. Primary cultured FTECs were also cryopreserved in liquid nitrogen for long-term storage.

### 4.2. Collection of Porcine Follicular Fluid (PFF)

The ovaries used for PFF collection were collected from healthy sows in the same manner as fallopian tubes. Ovaries were washed three times in PBS, and then, the PFF was collected only from large follicles by a vacuum pump, maintaining a constant pressure of 100 Pa. Subsequently, PFF was briefly centrifuged 10 min at 2000× *g* to remove any cells, then filtered with a 0.22 µm membrane. PFF was aliquoted and stored at −80 °C until use. PFF was diluted in DMEM/F12 according to the experiment’s purpose.

### 4.3. 3D and ALI Culture of FTECs

Expanded FTECs were resuspended with pre-cooled 100 µL Matrigel (#354234, Corning, Corning, NY, USA) and seeded onto a pre-warmed 24-well plate as a drop at a density of 25,000 cells per well. The Matrigel was solidified for 20 min at 37 °C; overlaid with 500 µL pre-warmed basal medium; and supplemented with E2, P4, or PFF. Three-dimensional organoid cultures were kept at 37 °C, 5% CO_2_ in a humidified incubator. The number and size of spheres with diameters > 50 μm were counted by ImageJ (Java 1.8.0_172).

### 4.4. Immunofluorescence Staining

Primary or differentiated FTECs were briefly washed with PBS and fixed in 4% paraformaldehyde (PFA, #A500684, Sangon Biotech, Shanghai, China) for 10 min at room temperature (RT), permeabilized, and blocked with 0.1% Triton X-100 (#9002-93- Sangon Biotech) and 5% fetal bovine serum (ABCONE, #B24726, Shanghai, China) for 30 min at RT. Cells were incubated with primary antibodies (Abs) at 4 °C overnight. The following primary Abs were used in this study: mouse anti-acetylated a-tubulin (1:500; #T6793, Sigma-Aldrich, St. Louis, MO, USA), rabbit anti-p73 (1:200; #ab40658, Abcam, Cambridge, UK). Alexa Fluor 488/568-conjugated secondary Abs (#A21202/A10042, Thermo Fisher Scientific, Carlsbad, CA, USA) were used at a dilution of 1:200 at RT for 1 h. Nuclei were counterstained with DAPI (#62248, Thermo Fisher Scientific). Finally, images were viewed and taken by a confocal microscope (Leica TCS SP8, Wetzlar, Germany) or fluorescent microscope (Nikon, ECLIPSE Ti2, Tokyo, Japan).

### 4.5. Western Blot Analysis

Cells were lysed with RIPA buffer (#89900, Thermo Fisher Scientific) containing a protease inhibitor cocktail (#04693159001, Sigma-Aldrich. The equivalent amounts of protein samples were separated by 8% SDS-polyacrylamide gel and then wet-transferred to a PVDF membrane at 200 mA and 4 °C for 2 h. The membrane was blocked using Tris-buffered saline with Tween 20 (TBS-T) containing 2% bovine serum albumin, followed by the incubation with primary Abs overnight at 4 °C followed by incubation with horseradish peroxidase-conjugated secondary Abs at RT for 1 h. Primary Abs were mouse monoclonal Abs against Foxj1 (1:2000, #ab235445, Abcam), Pax8 (1:2000, #10336-1-AP, Proteintech, Wuhan, China), and GAPDH (1:5000, #60004-1-IG, Proteintech). Secondary Abs were goat anti-rabbit IgG (#31460, Thermo Fisher Scientific) and goat anti-mouse IgG (#31430, Thermo Fisher Scientific). Signals were developed using an enhanced chemiluminescence substrate (#34580, Thermo Fisher Scientific), and images were acquired using an Azure Sapphire RGBNIR detection system.

### 4.6. Analysis of Proliferation Competency

The existence of precursor cells after inducing differentiation was confirmed by 5-Ethynyl-20-Deoxyuridine (EdU). Briefly, 5 μM EdU (#C10310-1, Ribobio, China) was added into the bottom chamber, and EdU could incorporate into proliferating cells during DNA synthesis. The detection of EdU was performed according to the manufacturer’s protocol.

### 4.7. TUNEL Assay

The PFF was added into the upper chamber to stimulate the differentiated FTECs cultured on transwells. After incubation, FTECs were washed briefly with PBS and fixed with 4% PFA. TUNEL staining was carried out using the in situ cell death detection kit (#11684795910, Roche) according to the manufacturer’s instructions.

### 4.8. Statistical Analysis

The results were expressed as mean ± standard deviation (SD) except otherwise stated. Student’s *t*-test was used to compare the variation between the two groups using GraphPad Prism software (Version 9.5.0). When comparing three or more groups, we analyzed the data with the ANOVA test. *p* values < 0.05 were considered statistical significance.

## Figures and Tables

**Figure 1 ijms-24-10154-f001:**
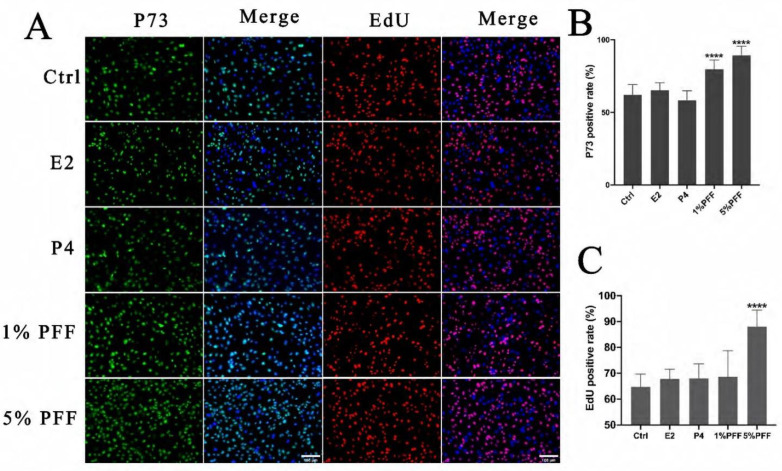
E2 is necessary and sufficient for ciliogenesis in FTECs: (**A**) Primary cultured FTECs were seeded and cultured with basal medium in the absence (Ctrl) and presence of 2 ng/mL E2, 10 nM P4, and 1%, 5% PFF for 24 h, respectively. The medium was changed after 24 h and 5 µM Edu was added to each group for 24 h. Then cells were fixed and stained with a basal cell marker (p73; red), and DAPI (blue)- and EdU-positive cells are shown in red. Scale bars: 100 µm. (**B**,**C**) Quantitation of p73- and EdU-positive cells presented in (**A**). (one-way ANOVA test, n = 7, compared with Ctrl group). Significance level: **** *p* < 0.0001.)

**Figure 2 ijms-24-10154-f002:**
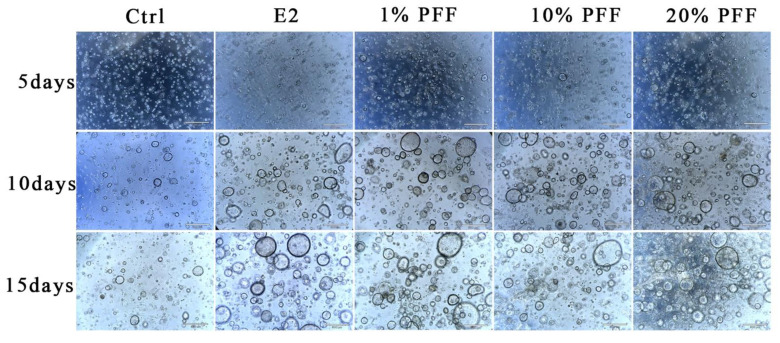
Phase contrast images of spheroid formation and growth: The organoids were cultured with basal medium and in the absence (Ctrl) and presence of 2 ng/mL E2, 10 nM P4, and PFF (1%, 10%, 20%), respectively. Small spheres are already visible 2 days after seeding and expand to reach a diameter of over 100 mm within 10 days. Images were taken after being cultured for 5, 10, and 15 days. Scale bars: 500 µm.

**Figure 3 ijms-24-10154-f003:**
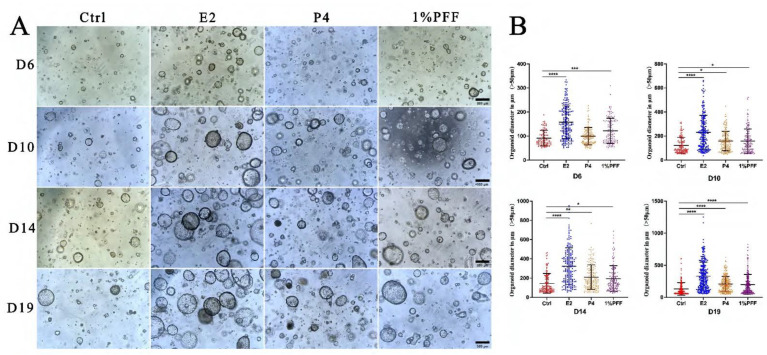
The hormones and PFF on the size and number of the organoid spheroids: (**A**) The organoids were cultured with basal medium in the absence (Ctrl) and presence of 2 ng/mL E2, 10 nM P4, and 1% PFF, respectively. Images were taken by a phase contrast microscope after being cultured for 6, 10, 14, and 19 days. The number and size of spheres with diameters greater than 50 μm were counted. Scale bars: 500 µm. (**B**) Quantitation of number and size of organoid spheres presented in (**A**). The data points represent the diameter of organoids measured in micrometers, and the median values with interquartile range are displayed for indicated culture conditions. Statistical significance was determined using one-way ANOVA tests and compared with control group. Significance level: * *p* < 0.05, ** *p* < 0.01, *** *p* < 0.001, and **** *p* < 0.0001.

**Figure 4 ijms-24-10154-f004:**
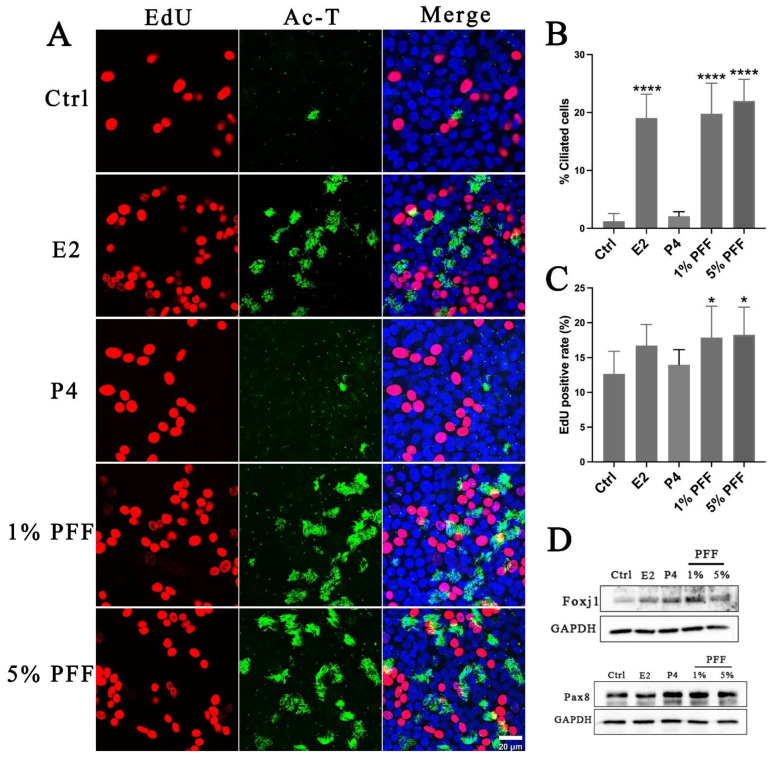
Effects of hormones and PFF on FTEC differentiation and self-renewal: (**A**) FTECs were cultured with the basal medium in the absence (Ctrl) and presence of E2 (2 ng/mL), P4 (10 nM), and PFF (1%, 5%), respectively, and exposed to ALI culture conditions for 10 days. EdU (5 µM) was added to basal chambers on ALI day 10. Differentiated cells were cultured with EdU for 48 h. Fixed cells were stained with anti-Ac-tubulin antibody (green) and DAPI (blue) and processed for EdU staining (red). Scale bars: 20 µm. (**B**) The numbers of ac-tubulin-positive cells were quantified (one-way ANOVA tests, n = 7, compared with Ctrl group), **** *p* < 0.0001). (**C**) The number of EdU-positive cells calculated in each medium (one-way ANOVA tests, n = 7, compared with Ctrl group). Significance level: * *p* < 0.05). (**D**). Western blot analysis of Foxj1 and Pax8 expression in FTECs after treatment by indicated hormones and PFF. FTECs were cultured with the basal medium in the absence (Ctrl) and presence of E2 (2 ng/mL), P4 (10 nM), and PFF (1%, 5%), respectively, and exposed to ALI culture conditions for 5 days.

**Figure 5 ijms-24-10154-f005:**
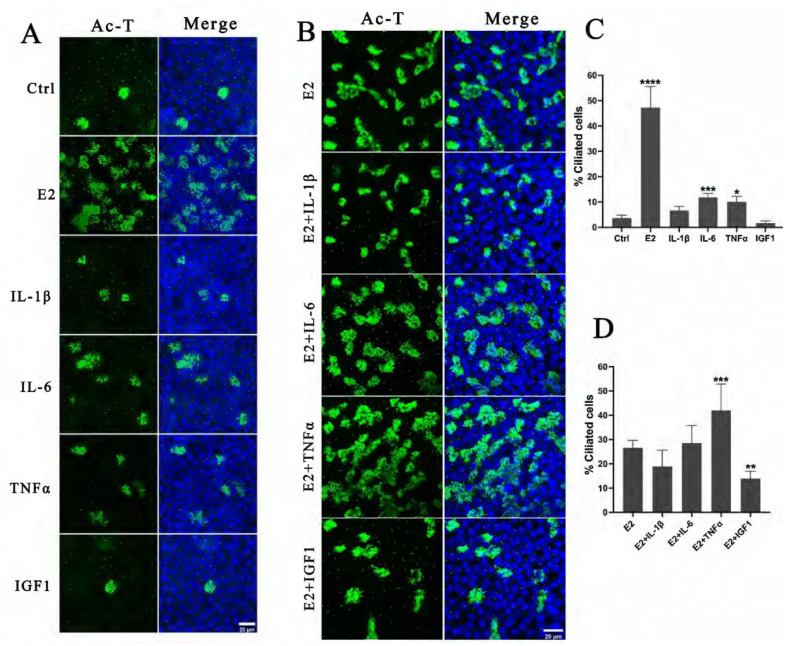
Effects of inflammation factors and growth factors existed in PFF on FTEC differentiation: (**A**) FTECs were incubated in basal medium (Ctrl) and basal medium supplied with E2 (2 ng/mL), IL-1β (10 ng/mL), IL-6 (10 ng/mL), TNFα (10 ng/mL), and IGF-1 (10 ng/mL), respectively, and exposed to ALI culture conditions for 10 days. Cells on ALI day 10 were stained for ac-tubulin (green) and nuclei (blue). (**B**) FTECs were incubated in basal medium supplied with E2 (Ctrl) or additionally supplied with IL-1β (10 ng/mL), IL-6 (10 ng/mL), TNFα (10 ng/mL), and IGF-1 (10 ng/mL), respectively; cells were exposed to ALI culture conditions for 10 days. (**C**) The numbers of ac-tubulin-positive cells in A were quantified (one-way ANOVA test, n = 7, compared with control group). (**D**) The numbers of ac-tubulin-positive cells in B were quantified (one-way ANOVA test, n = 7, compared with control group). Significance level: * *p* < 0.05, ** *p* < 0.01, and *** *p* < 0.001, **** *p* < 0.001.

**Figure 6 ijms-24-10154-f006:**
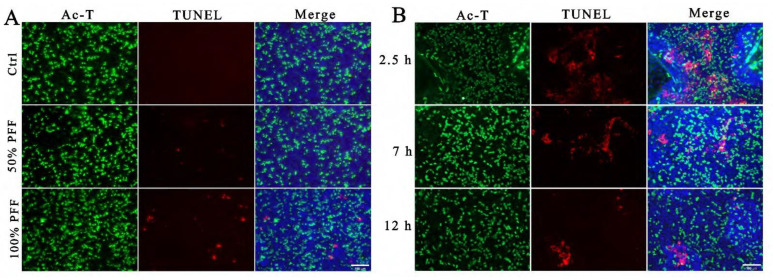
Simulating the direct stimulation of the oviduct epithelium by FF: FTECs were incubated in basal medium supplied with E2 (2 ng/mL) and exposed to ALI condition for 15 days. (**A**) Differentiated FTECs in the upper chambers were directly exposed to DMEM/F12, 50% PFF, or 100% PFF for 30 min; cells were then fixed and stained with anti-Ac-tubulin antibody (green) and DAPI (blue), and processed for TUNEL staining (red). Scale bars: 100 µm. (**B**) Differentiated FTECs in the upper chambers were directly exposed to 100% PFF for 2.5 h, 7 h, and 12 h, respectively. Cells were then fixed and stained with anti-Ac-tubulin antibody (green) and DAPI (blue) and processed for TUNEL staining (red). Scale bars: 100 µm.

## Data Availability

The data that support the findings of this study are available from the corresponding author upon reasonable request.

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
