# Peer review of "Effects of Follicular Fluid on Physiological Characteristics and Differentiation of Fallopian Tube Epithelial Cells Implicating for Ovarian Cancer Pathogenesis"

_ijms, 2023, doi:10.3390/ijms241210154_

Round 1

Reviewer 1 Report

The manuscript " Effects of follicular fluid on physiological characteristics and differentiation of fallopian tube epithelial cells implicating for ovarian cancer pathogenesis" by Zhu et al. reports a follicular fluid plays a role to promote fallopian tube epithelial cells in three different models including 2D, 3D, and ALI. The authors have compared the E2, P4, and FF on FTEC formation. Overall, this is a methodical, and well-organized manuscript that should be of interest to readers of IJMS. However, a few issues should be addressed before the manuscript is accepted for publication. The authors should address more about the relationship between tumor initiation and FF on FTEC formation. The authors should give a brief introduction and explanation about why 1%, 5%, 50%, or 100% PFF are chosen in this study. Figure 1b and c, Figure 4b and c, Figure 5c and d,  label the p-value in the correct form.   

Reviewer 2 Report

The distal end of FT is the most common original site of HGSC.But the mechanisms of the initiation of HGSC is unknown.None article investigate the mechanisms of the initiation of HGSC clearly, but this article illuminates some facts of this enigmatic mechanisms helping us prevent, diagnose and treat the ovarian cancer it in the future. In their works, they have established three FTECs culture models including 2D expanding culture, 3D, and ALI organoid culture model.They extensively investigated the effects of FF on the physiological functions, differentiation, and transformation of FTECs in different stage by using suitable culture model. This article can be better if the molecular mechanisms how hormones or other components of the FF influence the physiological characteristics and differentiation of FTEC. The article sets up several groups of controlled trials. I think it would be more beneficial to demonstrate whether the concentration of FF affects proliferation, differentiation, and transformation if the concentration interval of PFF increases, such as 1%,10%,50%,100%.In the last part of results, maybe you can add the ROS independently into the FTEC if the technology allows.    
